# Evaluation of Selected Medicinal, Timber and Ornamental Legume Species’ Seed Oils as Sources of Bioactive Lipophilic Compounds

**DOI:** 10.3390/molecules28103994

**Published:** 2023-05-09

**Authors:** Anna Grygier, Suryakant Chakradhari, Katarzyna Ratusz, Magdalena Rudzińska, Khageshwar Singh Patel, Danija Lazdiņa, Dalija Segliņa, Paweł Górnaś

**Affiliations:** 1Faculty of Food Science and Nutrition, Institute of Food Technology of Plant Origin, Poznań University of Life Sciences, Wojska Polskiego 31, 60-624 Poznań, Poland; 2School of Studies in Chemistry/Environmental Science, Pt. Ravishankar Shukla University, Raipur 492010, CG, India; 3Division of Fats and Oils Technology, Department of Food Technology, Institute of Food Science, Warsaw University of Life Sciences, Nowoursynowska 159c, 02-776 Warsaw, Poland; 4Department of Applied Sciences, Amity University, State Highway 9, Raipur Baloda-Bazar Road, Tilda, Raipur 493225, CG, India; 5Institute of Horticulture, Graudu 1, LV-3701 Dobele, Latvia

**Keywords:** Fabaceae, Leguminosae, phytostanol, bean, tocochromanol

## Abstract

Bioactive lipophilic compounds were investigated in 14 leguminous tree species of timber, agroforestry, medicinal or ornamental use but little industrial significance to elucidate their potential in food additive and supplement production. The tree species investigated were: *Acacia auriculiformis*, *Acacia concinna*, *Albizia lebbeck*, *Albizia odoratissima*, *Bauhinia racemosa*, *Cassia fistula*, *Dalbergia latifolia*, *Delonix regia*, *Entada phaseoloides*, *Hardwickia binata*, *Peltophorum pterocarpum*, *Senegalia catechu*, *Sesbania sesban* and *Vachellia nilotica*. The hexane-extracted oils of ripe seeds were chromatographically analysed for their fatty acid composition (GC-MS), tocochromanol (RP-HPLC/FLD), squalene and sterol (GC-FID) content. A spectrophotometrical method was used to determine total carotenoid content. The results showed generally low oil yield (1.75–17.53%); the highest was from *H. binata.* Linoleic acid constituted the largest proportion in all samples (40.78 to 62.28% of total fatty acids), followed by oleic (14.57–34.30%) and palmitic (5.14–23.04%) acid. The total tocochromanol content ranged from 100.3 to 367.6 mg 100 g^−1^ oil. *D. regia* was the richest and the only to contain significant amount of tocotrienols while other oils contained almost exclusively tocopherols, dominated by either α-tocopherol or γ-tocopherol. The total carotenoid content was highest in *A. auriculiformis* (23.77 mg 100 g^−1^), *S. sesban* (23.57 mg 100 g^−1^) and *A. odoratissima* (20.37 mg 100 g^−1^), and ranged from 0.7 to 23.7 mg 100 g^−1^ oil. The total sterol content ranged from 240.84 to 2543 mg 100 g^−1^; *A. concinna* seed oil was the richest by a wide margin; however, its oil yield was very low (1.75%). Either β-sitosterol or Δ5-stigmasterol dominated the sterol fraction. Only *C. fistula* oil contained a significant amount of squalene (303.1 mg 100 g^−1^) but was limited by the low oil yield as an industrial source of squalene. In conclusion, *A. auriculiformis* seeds may hold potential for the production of carotenoid-rich oil, and *H. binata* seed oil has relatively high yield and tocopherol content, marking it as a potential source of these compounds.

## 1. Introduction

Plant seeds contain various biologically active substances, including lipophilic substances such as phytosterols, tocochromanols and carotenoids, and are major sources of these micronutrients in the diet. Legume seeds tend to have low oil content; exceptions to this include soy, peanuts and *Pongamia pinnata*; however, low oil content does not exempt products as sources of valuable lipophilic compounds—wheat germ and rice bran contain very little oil, but are used for food supplement production. Legumes are also used in agronomical and agroforestry systems [1] as nitrogen fixers as well as grown for timber, firewood or ornamental purposes. In agroforestry, legumes can be either vegetable crops, such as beans, peas and lentils, cover or forage crops, such as alfalfa, or trees with wide canopies suitable for shading. In the meantime, seeds are produced in varying amounts by these trees, often used as food, and can supplement human diets not only with lean protein, but also with lipophilic bioactive compounds, potentially providing additional revenue to plantation managers. In the present study, the seeds of 14 timber, ornamental or medicinal legume species were analysed for their lipid profile.

It is important to note that both the species used in agroforestry and their purpose depends greatly on the climate zone. Nitrogen-fixing species are far more common and diverse in tropical and subtropical climate zones. In temperate and subtropical climate zones, trees are generally used for biomass. The other main use is for food. In tropical zones, their primary purpose is to increase crop harvests [2]. Additionally, forest-like plantations with low-intensity management can have higher biodiversity and carbon stocks than intensely managed monoculture plantations [3]. Of the investigated species, *Acacia auriculiformis*, *Albizia odoratissima*, *Delonix regia* and *Sesbania sesban* are already used (intentionally) or are present (naturally) in agroforestry systems [1,4,5]. Using N_2_-fixing trees poses several advantages. The direct effect is the supplementation of the soil with N_2_, leading to increased harvests in depleted soils. Additionally, trees provide shade to sensitive crops and windbreaks in erosion-prone areas. Their use is limited by the fixed N_2_ only becoming available once the leguminous species’ plant tissue decomposes. N_2_ fixation also varies greatly between species. Furthermore, it reduces or obviates the need for chemical fertilizers, reducing environmental pollution risk [6]. *Acacia auriculiformis* is mentioned most widely for its use in agroforestry as an intercropping or in silvopastures in humid to subhumid subtropics [5] alongside pineapple, papaya, curcumin, bananas, potatoes, *Colocasia alba* (elephant ear plant) and *Andrographis paniculata*, a medicinal plant [1]. *Sesbania* species, such as *S. sesban*, are suitable for improved fallows alongside a variety of crops in tropical climate zones at high elevation [7]. Other methods for enriching and improving nutrient availability in soils include supplementing the soil with natural fertilizers or microorganisms; for example, amending the soil with arbuscular mycorrhizal fungi and biochar also has a positive effect on growth [8].

*Dalbergia latifolia*, *Hardwickia binata*, *Peltophorum pterocarpum* and *Sesbania sesban* are used for timber, and the first two are considered premium hardwoods due to their colourful grain patterns, often referred to as rosewood [9]. Others are ornamental, such as *Brauhinia racemosa*, *Cassia fistula*, *Delonix regia* and aforementioned timber species *A. auriculiformis* and *P. pterocarpum*. Several of the species are used in traditional medicines and hair cosmetics (*Acacia concinna* and *Entada phaseoloides*) due to the presence of saponins; however, used plant parts and purposes vary between regions [5]. Some species’ leaves or pods are gathered for animal fodder, such as those of *Albizia lebbeck* and *P. pterocarpum* [10].

Plant parts of several species are used in food, such as the fresh leaves of *A. auriculiformis*, flowers and young leaves of *A. concinna* or flowers of *C. fistula*. Only a couple of species’ fruit are eaten: *A. concinna* (roasted seeds), *S. catechu* (seeds) and *V. nilotica* (fruit pulp). Sap gums of *V. nilotica* (gum arabica) and *A. auriculiformis* are also produced, though only the former is produced widely. While various plant parts are gathered and used, none are grown or harvested for the fruit on a significant scale; therefore, global cultivation area and potential harvest are not known.

While many species’ leaves and seeds are gathered for animal fodder, few are used in food. One of the reasons for this are various antinutritional factors present in legume seeds, including high tannin content, trypsin inhibitors, phytic acid and saponins, although these are not equally dispersed in seed components. For example, in *S. sesban* seeds, the majority of tannins are located in the seed coat and endosperm while other antinutritional factors (phytic acid, trypsin inhibitors and lectin) are generally concentrated in the cotyledon and saponin concentration is similar in the two fractions [11]. An additional problem with legume seeds as food is the presence of toxins in some species, including cyanogenic compounds and nonproteinogenic amino acids. Although there are no reports on specific toxins in the investigated species, there are reports of albizziin and *S*-(β-carboxyethyl)-cysteine in *Albizia julibrissin* [12], and *A. lebbeck* has shown toxicity in mice [13]. The compounds responsible for antinutritional factors and toxicity of legume seeds are water-soluble and, therefore, of less concern if the seeds are used for oil extraction.

Previous investigations of these species have considered them both as sources of oil for biodiesel production, raw material for solid fuels or assistants in production of fuels. For example, *A. nilotica* (*V. nilotica*) was tested as an oil source for biodiesel [14], producing an AOCS-conformant product. On the other hand, pods of low-fat legumes, such as *A. lebbeck*, have been tested as biocatalysts in the production of biodiesel [15]. In such studies on biodiesel, minor lipophilic compounds are of little concern and rarely analysed. Although the seed oil content and fatty acid composition of most species is well documented, information on the lipophilic micronutrient content is often sparse unless the species is widely cultivated as a food crop. In other cases, even fatty acid composition is not known. To the best of our knowledge, this study includes the first investigation of *D. latifolia* fatty acid composition and lipophilic profile.

Phytosterols are steroids, similar to cholesterol found in plants, that serve as structural membrane components, while stanols are saturated compounds with similar structures. Supplementing the diet with phytosterols can reduce blood cholesterol, cardiovascular disease risk and inflammation and modulate immune responses in asthma patients [16]. The main sources of phytosterols and stanols are cereals, cooking fats and oils [17]. Significant improvements are seen if at least 2 g of phytosterols are ingested per day, though smaller intakes have also demonstrated positive effects [18]; typical daily intake is around 200–320 mg [17]. Phytosterol content in oils typically ranges from 150 mg 100 g^−1^ in palm oil to 893 mg 100 g^−1^ in canola and 990 mg 100 g^−1^ in corn oil, but wheat germ (967 mg 100 g^−1^) and rice bran (1891 mg 100 g^−1^) oil are regarded as especially rich [19,20]. Legumes are not a major contributor to phytosterol intake [17]. Certain legume seed oils, such as *T. indica* (tamarind), are both rich in phytosterols and have relatively good oil yield [21] but are not produced commercially.

Carotenoids are naturally occurring yellow, orange and red lipophilic pigments with antioxidant properties. They have various bioactive functions in the body, including maintaining eye health and cognitive function, and β-carotene is a provitamin of vitamin A. Vegetables and leafy greens are generally regarded as the best sources of carotenoids [22], as oils typically have quite low carotenoid content—canola oil contains around 4.5 mg 100 g^−1^ [23]. Generally, the major carotenoids in legume seeds are all-*trans*-lutein and all-*trans*-zeaxanthin, while β-carotene content is small, and the total content and composition of carotenoids in legume seeds is quite variable [24,25].

Tocochromanols are a group of naturally occurring prenyllipids consisting of a chromanol ring and prenyl tail that can have unsaturated bonds. The chromanol ring can contain one, two or no methyl groups. The most common examples are tocopherols (Ts) and tocotrienols (T3s). Tocochromanols act as lipophilic antioxidants and some have vitamin E activity. Commonly used oils typically contain between 18 mg 100 g^−1^ (olive oil) and 60 to 70 mg 100 g^−1^ (canola, sunflower and corn oil), and wheat germ oil is particularly rich (257 mg 100 g^−1^) [26]. Legumes are not major tocochromanol sources and typically contain either α- or γ-T in significant concentration, while tocotrienols are either not present or are very minor components [21,27,28,29].

## 2. Results and Discussion

Fourteen species’ (some common names provided in parentheses)—*Acacia auriculiformis* (auri), *Senegalia catechu* = *Acacia catechu* (catechu), *Acacia concinna* = *Senegalia rugata* (shikakai), *Vachellia nilotica* = *Acacia nilotica* (gum arabic tree), *Albizia lebbeck* (siris, sirisa), *Albizia odoratissima* (Ceylon rosewood, black siris), *Bauhinia racemosa* (bidi leaf tree), *Cassia fistula* (golden shower), *Dalbergia latifolia* (Indian rosewood), *Delonix regia* (royal poinciana), *Entada phaseoloides* (box bean), *Hardwickia binata* (anjan), *Peltophorum pterocarpum* (copperpod) and *Sesbania sesban* (Egyptian riverhemp, sesban)—were analysed for their lipophilic profile: fatty acid composition, tocochromanol (tocopherol and tocotrienol), squalene and sterol content as well as total carotenoid content.

### 2.1. Oil Yield and Fatty Acid Profile

Most of the species provided low or very low oil yield (1.7% in *A. concinna* to 8.3% in *A. lebbeck*), but *D. latifolia* (10.8%), *A. auriculiformis* (13.4%) and *H. binata* (17.5%) exceeded 10% oil yield. The oil content does not qualify any of the tested species as viable oilseeds or raw material for biodiesel production, although the fatty acid make-up is favourable. The oil yields are lower than observed in previous reports, for example, Adweuyi et al. [30] produced a 7.0% and 8.1% oil yield from *D. regia* and *P. pterocarpum*, while only 3.1% and 6.5% were produced in the present study. There are two possible explanations for the lower oil yield: lower fat content in the seeds used for analysis or lower extraction efficiency and recovery of the method used for oil extraction. Fat content varies between individuals and can be affected by climatic and abiotic factors [31]—salt, drought and other stresses hinder photosynthesis, resulting in lower fat content as well. Oil yield from analysed samples is shown in Figure 1.

Considering the low oil yield from the seeds, effective use of the material is an important consideration. When processed, legumes are generally used to produce protein or carbohydrate-based products. The oil extraction method used for defatting plays a key role in the properties of hydrophilic extractables such as protein [32]. Ultrasonication was used in the present study to improve oil extraction efficiency but it can also be used to improve the properties of dough [33] and protein isolates [34].

Fatty acid composition (Table 1 and Table 2) was largely in agreement with previous investigations where such data was available. In most of the investigated oils, linoleic (42.34–62.28%), oleic (13.62–32.89%) and palmitic acid (5.14–23.04%) dominated, with other fatty acids generally constituting minor fractions. The highest proportion of linoleic acid was observed in *A. odoratissima*, at 62.28%, along with a relatively high proportion of linolenic acid (C18:3; 2.01%), resulting in the highest proportion of polyunsaturated fatty acids (PUFA), 64.29%. Unlike previous investigations, *V. nilotica* oil contained mainly linoleic acid (52.39%) and oleic acid (26.20%). A previous investigation [14] of chloroform hexane-extracted oil observed oleic (37.27%) acid as the dominant fatty acid, followed by linoleic (31.01%) and a much higher proportion of palmitic acid (18.45%). Similarly, the present study observed a different fatty acid proportion in *C. fistula* than a previous study: although linoleic acid (54.51%) constitutes a similar proportion to that in [35], the present study observed a much higher proportion of oleic acid (34.21%) and a much lower proportion of palmitic acid (8.51%) than the 19.8% and 20.1%, respectively, observed previously. The fatty acid profile of *D. regia* and *P. pterocarpum* are similar to previous investigations [30].

Although, like the other investigated oils, *D latifolia* seed oil contained predominantly linoleic and oleic acid, the proportion of linolenic acid was much higher (2.35%) than most of the other samples. An exception in regard to fatty acid composition is *H. binata*: larger proportions of behenic (C22:0; 15.7%) and lignoceric (C24:0; 11.38%) acid were present, which have not been observed in previous investigations. Since these very-long-chain-fatty acids have not been observed in the species’ seeds before, the differences could be a result of various factors during seed development.

The fatty acid make-up of *S. catechu* was similar to a previous report [36] in regard to most FAs, differing only in the lower proportion of palmitic acid. The trend is continued with *S. sesban*—previous papers [11] reported a higher proportion of palmitic acid (13.3–14.1%) than the 8.56% observed in this study. Hossain and Becker [11] also reported a lower proportion of oleic acid—only 10.3 to 13.2%. Differences in fat content and fatty acid make-up can be a result of many abiotic and biotic factors [31,37,38], including genetic predisposition, seed maturity and extraction method, though the choice of solvent only has a slight effect on the proportion of fatty acids. For example, salt stress affects overall photosynthesis while drought stress can result in significantly lower oil content as well as a different, decreased oleic acid content [31].

### 2.2. Tocochromanols and Total Carotenoids

The total carotenoid content ranged from 0.7 mg 100 g^−1^ in *E. phaseoloides* to 23.7 mg 100 g^−1^ in *A*. *auriculiformis* (Table 3). This is within the range of total carotenoid content (0.7–43.1 mg 100 g^−1^) previously observed in legume seeds [28,29].

Some of the oils had exceptionally high total carotenoid content: *A. auriculiformis* (23.7 mg 100 g^−1^), *S. sesban* (23.6 mg 100 g^−1^), *A. odoratissima* (20.4 mg 100 g^−1^) and *A. lebbeck* (18.6 mg 100 g^−1^). Previous investigation of *D. regia* petals found a higher (25.2 mg 100 g^−1^) concentration of carotenoids [39], but no investigation of the seeds has been done. Of the analysed oils, the simultaneous high fat and carotenoid content of *A. auriculiformis* make it a prospective source of carotenoid-rich oil, although more detailed analysis of the constituent carotenoid compounds is needed to evaluate a safe intake of the oil. Since carotenoids were determined using spectrophotometrical methods, the actual carotenoid content could be lower or higher due to the varying absorbance of different carotenoid molecules at a given wavelength. Additionally, the results do not provide information on the make-up of the oil carotenoids. All-*trans*-lutein and All-*trans*-zeaxanthin [24,25] have been observed as the main carotenoid in common edible pulses but may be not dominant in the analysed oils. An important consideration in producing carotenoids from natural sources is their availability. Seeds can be harvested once a year; the harvest is highly dependent on climatic conditions and susceptible to pests and diseases. Considering that carotenoids, such as lycopene, can be effectively produced by genetically engineered photosynthetic microorganisms in bioreactors [40], the cost of availability is a serious concern.

Tocopherols (Ts) were dominant in the oils; the total tocochromanol content ranged from 100.3 to 367.5 mg 100 g^−1^, which is within the range (71.9 to 444.8 mg 100 g^−1^ oil) previously observed in other legume species’ seed oils [21,28,29]. The highest tocochromanol content was observed in *D. regia*, which contained tocotrienols (5.7% of total tocochromanols), typically not present in legume seed oils even when the total tocochromanol content is high [29]. Tocotrienols either were not present in any of the other investigated oils or were found only in negligible amounts. A relatively high concentration has previously only been observed in *Pentaclethra filamentosa* = *P. macroloba* by [41]; however, tocotrienol standards are not listed in used materials and it is unclear what method was used to confirm the identity of the compounds. As such, the notable concentration of tocotrienols in *D. regia* oil should be regarded with caution and more studies are needed to corroborate this finding. Legume oils contain either α-T or γ-T as the largest fraction across a wide range of tocochromanol concentrations [21,27,28,29]. The other oils contained mainly α- and γ-T in varying proportion, with most of the oils containing more α-T than γ-T except for *D. latifolia*, which contained almost exclusively γ-T. A previous investigation of *A. lebbeck* seed oil reported a much smaller tocopherol concentration (84.93 mg 100 g^−1^) and a different tocopherol profile. The proportion of α-T was larger, the γ-T was smaller and a higher β-T content and proportion was observed [42].

The differences observed in tocochromanol content can be attributed to a different oil extraction protocol [43]; tocochromanol content tends to increase in cases of drought stress [31].

A previous study of *C. fistula* seed oil also reported slightly different results: lower total tocochromanol concentration and a slightly different profile [44]. While tocopherols were present in similar proportion, it also reported the presence of β-, α- and γ-tocotrienol, none of which were observed in the present study. Conversely, the present study only noted a small concentration of δ-tocotrienol, which was not observed at all by Phuong et al. [44]. Discrepancies can be explained both by different extraction and tocochromanol determination methods and different actual tocochromanol contents in oil samples as a result of biotic and abiotic factors during plant growth and seed development.

Although the total tocochromanol content in *D. regia* oil is high, it is limited as a source of tocochromanols by the low oil yield (3.1%), as is the case for the rest of the investigated oils. Taking oil yield into account, *H. binata* and *D. latifolia* make the most efficient sources of tocochromanols.

### 2.3. Sterols and Squalene

The sterol and squalene content of analysed seed oils is provided in Table 4 and total sterol content in depicted in Figure 2. In almost all of the oils, β-sitosterol dominated, except for *S. catechu*, in which Δ5-stigmasterol was dominant, and *S. sesban*, which contained an even concentration of the two. The oils contained other sterols in comparatively minor proportions. The total sterol content was similar or lower to that in widely consumed vegetable oils [20], apart from a few: *A. concinna* (2543 mg 100 g^−1^), *C. fistula* (1400 mg 100 g^−1^), *S. catechu* (1350 mg 100 g^−1^) and *D. regia* (1271 mg 100 g^−1^). All of these, however, have a low fat content, with *S. catechu* possessing the highest at 5.6%.

In most of the oils, β-sitosterol constituted the absolute largest portion, with other sterols at only minor concentration, with some exceptions. Following β-sitosterol, campesterol, Δ5-stigmasterol and Δ7-stigmasterol were present in comparatively higher concentration. Oils which did not predominantly contain β-sitosterol include *S. catechu*, in which Δ5-stigmasterol was dominant and campesterol had a significant proportion as well; *P. pterocarpum*, the seed oil of which contained very little β-sitosterol and contained more Δ5-stigmasterol than any other sterol; and *S. sesban*, which contained β-sitosterol and Δ5-stigmasterol in equal proportions.

Few investigations of the sterol content in these seed oils have been carried out. The presence, but not concentration, of stigmasterol has been observed in *D. regia* and *P. pterocarpum*, and sitosterol has been reported in *D. regia* seed oil [30], confirmed in the present study. Quantitative analyses have previously been performed on *C. fistula* seed oil, reporting β-sitosterol at 62% of total sterols, followed by stigmasterol and campesterol at much lower levels [44].

Although the present study also identified β-sitosterol as the main sterol, only Δ5-stigmasterol was observed as another major sterol component. Additionally, γ-sitosterol has been isolated from *A. nilotica* (syn. *V. nilotica*) leaves [45], but was not observed in the present study.

Significant squalene content was only observed in *C. fistula*, but it was lower than in olive or rice bran oil and would not provide a more efficient source of squalene. Although squalene has not been analysed in the seeds before, absence or low concentration is common for legume seeds [21,28,29].

## 3. Materials and Methods

### 3.1. Plant Material

The seeds of fourteen legume species (some common names provided in parentheses) were analysed: *Acacia auriculiformis* (auri), *Senegalia catechu* = *Acacia catechu* (catechu), *Acacia concinna* = *Senegalia rugata* (shikakai), Vachellia nilotica = *Acacia nilotica* (gum arabic tree), *Albizia lebbeck* (siris, sirisa), *Albizia odoratissima* (Ceylon rosewood, black siris), *Bauhinia racemosa* (bidi-leaf tree), *Cassia fistula* (golden shower), *Dalbergia latifolia* (Indian rosewood), *Delonix regia* (royal poinciana), *Entada phaseoloides* (box bean), *Hardwickia binata* (anjan), *Peltophorum pterocarpum* (copperpod) and *Sesbania sesban* (Egyptian riverhemp, sesban). The seeds were collected in March–April 2017 on Pt. Ravishankar Shukla University campus in Amanaka, Raipur, Chhattisgarh, India (21°14′31.2″ N 81°35′21.5″ E). Species were authenticated by prof. Khageshwar Singh Patel using a standard monograph [46] and the National Botanical Research Institute (Lucknow, India). The pictures of seeds can be found in Appendix A. Drying was carried out in open airflow in 30 ± 10 g batches until less than 10% moisture was reached. Dried samples were stored at −18 ± 1 °C for 1 to 3 months until further analysis. Upon analysis, residual moisture was removed by freeze-drying whole seeds (FreeZone, Labconco, Kansas City, MO, USA), which were then powdered in a MM 400 mixer mill (Retsch, Haan, Germany) and immediately extracted. Dry mass was determined gravimetrically. All seed oil analyses were performed in 2017–2018. The species’ uses are provided in Table 5 according to mentions in the scientific literature.

### 3.2. Solvents, Standards and Reagents

The mixture of thirty-seven fatty acid methyl esters (FAME) and *N*,*O*-bis-(trimethylsilyl)-trifluoroacetamide (BSTFA) with 1% trimethylchlorosilane (TMCS) were purchased from Supelco (Steinheim, Germany).

Four homologues (α, β, γ and δ) each of tocopherol and tocotrienol HPLC-grade standards (>95% purity) were used for tocochromanol identification and quantification. Tocopherol standards were purchased from Merck (Darmstadt, Germany) and tocotrienol from LGC Standards (Teddington, Middlesex, UK).

High performance liquid chromatography (HPLC) grade solvents, including methanol, 2-propanol, *tert*-butyl methyl ether, *n*-hexane, and gas chromatography (GC)-grade standards (≥97% 5α-cholestane, ≥95% brassicasterol, campesterol, stigmasterol, β-sitosterol, cholesterol, squalene) for sterol and squalene analysis were purchased, as well as any other unlisted solvents, reagents and standards, from Sigma-Aldrich (Steinheim, Germany).

### 3.3. Oil Extraction Using Ultrasound-Assisted Extraction

Oils were extracted using an organic solvent (*n*-hexane) according to Cravotto et al. [56] and Górnaś et al. [57], as the method provides a higher oil yield in comparison to conventional methods. In brief, 5 g of ground legume seeds were vortexed with 25 mL of *n*-hexane in a centrifuge tube on a Vortex REAX top (Heidolph, Schwabach, Germany) at 2500 rpm for 1 min. Then, samples were subjected to ultrasound treatment in the Sonorex RK 510 H ultrasonic bath (Bandelin electronic, Berlin, Germany) for 5 min at 35 °C. The ultrasound treatment allows for the increase of extraction efficiency. The samples were then centrifuged (10,000× *g* for 5 min at 21 °C) in a Centrifuge 5804 R (Eppendorf, Hamburg, Germany). The supernatant was decanted into a round-bottom flask and the remaining solid residue was re-extracted twice using the procedure described above. The supernatants collected from all 3 extractions were evaporated at 40 °C using a Laborota 4000 vacuum rotary evaporator (Heidolph, Schwabach, Germany) until a constant mass was reached.

Extracted oil was stored at −18 ± 1 °C until further analysis for 1 to 3 months. The following formula was used to calculate oil yield:$$Oil\;yield,\;\%\;dm=\frac{W1}{W2}\times 100\%$$
where *W*1 = oil mass after solvent evaporation, g; *W*2 = dry seed mass used for oil extraction, g.

### 3.4. Fatty Acid Analysis

Sample fatty acids were first esterified into FAMEs, following standard AOCS (American Oil Chemists’ Society) protocol [58], and then analysed according to a method developed by Górnaś et al. [59] in a GC system (Thermo 1300, Thermo Scientific, Waltham, MA, USA) equipped with an SP^TM^-2560 capillary column (100 m × 0.25 mm × 0.2 µm) (Supelco, Bellefonte, PA, USA) and flame ionization detector (FID), starting at 160 °C, increasing the temperature by 6 °C per minute until it reaches 220 °C and then finally holding this temperature for 17 min. Injection port temperature was set to 240 °C, hydrogen was used as a carrier gas (flow rate 1.5 mL min^−1^). Results were expressed as % of the individual FAME relative to the total FAME peak area in a sample chromatogram. To confirm the identified FAMEs, the GC system was coupled with an Agilent 7000 Triple Quad mass spectrometer (MS) (Agilent, Santa Clara, CA, USA) with a Supelcowax-10 (30 m × 0.25 mm × 0.5 µm) column (Supelco, Bellefont, PA, USA) using helium as the carrier gas (flow rate 34.6 mL s^−1^). The initial oven temperature was held at 40 °C for 1 min, then increased by 5 °C/min until it reached 220 °C and held at 220 °C for 30 min. Injector temperature was set at 220 °C, split ratio 50:1. Electron impact mode (70 eV) was used to record mass spectra, scanning at 33–330 *m*/*z* range. Ion source temperature was 230 °C, with scan time 100 and 0.1 MS step size.

### 3.5. Total Carotenoid Analysis

A spectrophotometrical analysis protocol, as provided by Górnaś et al. [60], was used to determine total carotenoid content. Oils (0.2 g) were diluted with 5 mL *n*-hexane in volumetric flasks and absorbance was measured in a UV-1800 spectrophotometer (Shimadzu, Kyoto, Japan) at 450 nm. Results were calculated using (all-*E*)-β-carotene molar extinction coefficient (ε = 139,048) and the following Beer–Lambert law-compliant equations were used to express molar concentration as (all-*E*)-β-carotene equivalent:$$C=\frac{A}{\varepsilon \times 1}$$
m=c×MW×V
where *C* = carotenoid concentration, mol/L; *A* = absorbance at 450 nm; *ε* = molar extinction coefficient, L mol^−1^ cm^−1^; 1 = pathlength/cuvette width; *m* = total carotenoid mass in 1 g of the sample; *MW* = molecular weight, g mol^−1^; *V* = solution volume, L.

### 3.6. Tocochromanol (Tocopherol and Tocotrienol) Analysis

Sample preparation was carried out according to the method used by Górnaś et al. [61]. Seed oil samples were mixed with 2-propanol at a 1:100 (*v*/*v*) ratio. Tocochromanol content was determined in an RP-HPLC/FLD system (Shimadzu, Kyoto, Japan) consisting of a pump (LC-10ADvp), a degasser (DGU-14A), a low pressure gradient unit (FCV-10ALvp), a system controller (SCL-10Avp), an auto injector (SIL-10AF), a column oven (CTO-10ASvp) and a fluorescence detector (RF-10AXL) and equipped with a guard (4 × 3 mm) and Luna pentafluorphenyl (PFP) column (4 × 3 mm and 150 × 4.6 mm, respectively, 3 μm) (Phenomenex, Torrance, CA, USA), capable of separating all tocochromanol isomers (β and γ), according to a validated method used by Górnaś et al. [62]. Authentic standards were used in expressing results. Tocochromanol analysis was performed in isocratic methanol:water (93:7) flow (1.0 mL min^−1^) at 40 °C; ambient temperature set at 22 ± 1 °C and analysis run time was 13 min. Calibration curves established previously for each tocopherol and tocotrienol homologue standard using a fluorescence detector (excitation wavelength at 295 nm, emission at 330 nm) were applied for the identification and quantification of tocochromanols. Tocopherols (Ts) and tocotrienols (T3s) had the following limits of detection (LODs): 0.051 (α-T), 0.018 (β-T), 0.022 (γ-T), 0.044 (δ-T), 0.061 (α-T3), 0.027 (β-T3), 0.030 (γ-T3) and 0.019 (δ-T3) mg mL^−1^.

### 3.7. Sterol and Squalene Analysis

Sample saponification and silylation was carried out according to the AOCS protocol [63]. The oil sample (50 mg) was saponified with 1 M KOH in methanol for 18 h at room temperature. Unsaponifiables were then extracted thrice with *n*-hexane:*tert*-butyl methyl ether (1:1, *v*/*v*). Silylated with BSTFA + 1% TMCS, phytosterols, cholesterol and squalene were separated in a HP 6890 gas chromatograph (Hewlett Packard, Wilmington, DE, USA) equipped with a DB-35MS capillary column (25 m × 0.20 mm × 0.33 μm; J&W Scientific, Folsom, CA, USA) and flame ionization detector (FID). In splitless mode, 0.5 μL of the sample were injected into the column with hydrogen as the carrier gas (flow rate 1.5 mL min^−1^) with initial column temperature set at 100 °C for 5 min, then increased by 25 °C min^−1^ until it reached 290 °C, held for 1 min, then increased to 290 °C at 3 °C min^−1^ and held for 20 min. Injection port temperature was held at 290 °C and FID detector was held at 300 °C. Chromatographic conditions were chosen according to Górnaś et al. [60]. An internal standard (5α-cholestane) was used to calculate phytosterol, cholesterol and squalene concentration. The LOD was 0.01 μg g^−1^ for all analytes. Campesterol, stigmasterol, β-sitosterol, cholesterol and squalene were identified by comparison with retention time of standards. Other phytosterols were identified on an MS system due to lack of commercial standards on the market. The same chromatographic conditions were used when running GC-MS as described above for GC-FID. Electron impact mode (70 eV) was used when gathering mass spectra, scanning masses in the 100 to 600 Da range. Both systems, GC-FID and GC-MS, were calibrated before sample analysis using commercial standards (brassicasterol, campesterol, stigmasterol, β-sitosterol, cholesterol and squalene). Kováts retention indices were calculated between GC-FID and GC-MS to confirm all sterols were identified correctly.

### 3.8. Statistical Analysis

The results were presented as means ± standard deviation from three different replications (*n* = 3) from independent batches (each batch was selected randomly from harvested seeds). Statistically significant differences between samples were determined using one-way analysis of variance (ANOVA) (*p* < 0.05) along with the Bonferroni post hoc test. Statistical analysis was carried out using Statistica 10.0 (StatSoft, Tulsa, OK, USA).

## 4. Conclusions

The analysed oils contained mainly linoleic acid and were generally highly unsaturated, except for *H. binata* and *P. pterocarpum*, which were uncharacteristically rich in long-chain saturated fatty acids. Although several of the investigated oils were rich is tocopherols, carotenoids and sterols, their use as sources of bioactive compound-rich oils is limited by low oil yield as well as lack of industrial harvesting and processing. Of the tested legume seeds, *A. auriculiformis* can be considered as a raw material for the production of carotenoid-rich oil because of the relatively high oil yield and frequent use of the plant in agroforestry systems. It is important to identify the carotenoid compounds present in the oil before advising it as a supplement or source of lipophilic antioxidants, as the biological and antioxidant activity of different carotenoids is not equal. Additionally, *H. binata* seed oil had good yield and a relatively high tocopherol content, especially of α-tocopherol, although it was not the sole tocopherol present.

Considering the low or relatively low oil yield from all tested species’ seeds, additional research on the properties of the hydrophilic (protein and carbohydrate) levels are also advisable to fully understand the economic and nutritional potential of these plants.

## Figures and Tables

**Figure 1 molecules-28-03994-f001:**
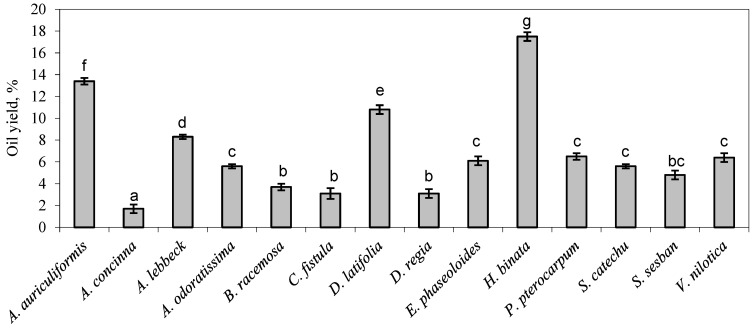
Legume seed oil yield using ultrasound-assisted solvent extraction. Different numbers indicate statistically significant differences at *p* ≤ 0.05.

**Figure 2 molecules-28-03994-f002:**
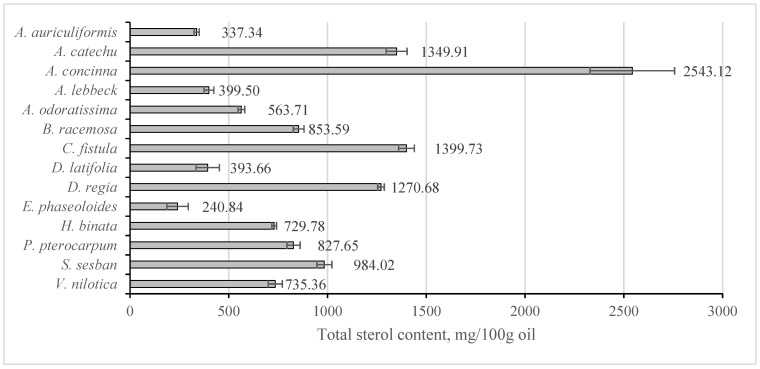
Total sterol content (mg 100 g^−1^ oil) in the investigated legume seed oils.

**Table 1 molecules-28-03994-t001:** Saturated fatty acid composition (% of total fatty acids) in the investigated legume seed oils.

Species	Fatty Acid					
	C14:0	C16:0	C18:0	C20:0	C22:0	C24:0
*A. auriculiformis*	nd	12.67 ± *0.55* ^e^	1.10 ± *0.08* ^a^	0.81 ± *0.08* ^b^	3.72 ± *0.18* ^e^	0.64 ± *0.04* ^b^
*A. concinna*	0.04 ± *0.01* ^a^	9.27 ± *0.10* ^c^	1.06 ± *0.04* ^a^	0.84 ± *0.06* ^b^	0.12 ± *0.02* ^a^	nd
*A. lebbeck*	0.09 ± *0.01* ^b^	14.03 ± *0.44* ^f^	5.90 ± *0.12* ^e^	4.44 ± *0.13* ^f^	5.09 ± *0.20* ^f^	0.43 ± *0.02* ^a^
*A. odoratissima*	nd	11.95 ± *0.41* ^de^	3.78 ± *0.02* ^d^	1.65 ± *0.13* ^de^	3.95 ± *0.10* ^e^	nd
*B. racemosa*	nd	8.98 ± *0.18* ^bc^	1.04 ± *0.09* ^a^	0.87 ± *0.02* ^b^	0.16 ± *0.01* ^a^	nd
*C. fistula*	0.04 ± *0.01* ^a^	8.51 ± *0.27* ^b^	0.93 ± *0.02* ^a^	0.75 ± *0.03* ^ab^	0.17 ± *0.02* ^a^	nd
*D. latifolia*	0.40 ± *0.02* ^c^	15.34 ± *0.52* ^g^	6.28 ± *0.20* ^e^	1.89 ± *0.06* ^e^	1.34 ± *0.05* ^d^	1.77 ± *0.07* ^d^
*D. regia*	nd	16.07 ± *0.24* ^g^	10.17 ± *0.23* ^g^	0.89 ± *0.03* ^bc^	0.37 ± *0.02* ^b^	nd
*E. phaseoloides*	0.03 ± *0.01* ^a^	8.23 ± *0.34* ^b^	4.08 ± *0.18* ^d^	1.27 ± *0.08* ^d^	0.69 ± *0.03* ^c^	0.61 ± *0.05* ^b^
*H. binata*	nd	5.14 ± *0.20* ^a^	1.87 ± *0.07* ^b^	1.98 ± *0.08* ^e^	15.70 ± *0.58* ^g^	11.38 ± *0.30* ^e^
*P. pterocarpum*	nd	23.04 ± *0.41* ^i^	9.16 ± *0.28* ^f^	0.64 ± *0.04* ^a^	0.21 ± *0.02* ^a^	nd
*S. catechu*	0.10 ± *0.01* ^b^	17.97 ± *0.49* ^h^	2.70 ± *0.11* ^c^	1.29 ± *0.06* ^d^	1.28 ± *0.05* ^d^	0.96 ± *0.04* ^c^
*S. sesban*	0.07 ± *0.02* ^ab^	8.56 ± *0.17* ^b^	1.01 ± *0.22* ^a^	0.84 ± *0.03* ^b^	0.14 ± *0.02* ^a^	nd
*V. nilotica*	0.10 ± *0.02* ^b^	11.72 ± *0.28* ^d^	6.42 ± *0.21* ^e^	0.93 ± *0.04* ^c^	0.99 ± *0.04* ^d^	nd

Values are expressed as the mean ± *standard deviation* (italic values) (*n* = 3). Different letters in the same column indicate statistically significant differences at *p* ≤ 0.05; nd, not detected.

**Table 2 molecules-28-03994-t002:** Unsaturated fatty acid composition (% of total fatty acids) in the investigated legume seed oils.

Species	Fatty Acid					
	C16:1	C18:1	C18:2	C18:3 *n*-3	C20:1	C20:2
*A. auriculiformis*	0.41 ± *0.02* ^b^	23.89 ± *0.42* ^cd^	55.09 ^de^ ± *1.28*	0.38 ± *0.08* ^ab^	0.36 ± *0.05* ^a^	0.12 ± *0.02* ^b^
*A. concinna*	0.05 ± *0.02* ^a^	32.58 ± *0.33* ^e^	55.20 ^d^ ± *0.38*	0.41 ± *0.03* ^b^	0.35 ± *0.02* ^a^	0.07 ± *0.01* ^a^
*A. lebbeck*	0.34 ± *0.01* ^b^	22.12 ± *0.14* ^c^	46.26 ^b^ ± *1.03*	0.53 ^c^ ± *0.02*	0.48 ± *0.02* ^b^	0.07 ± *0.00* ^a^
*A. odoratissima*	0.76 ± *0.02* ^d^	13.62 ± *0.23* ^a^	62.28 ^g^ ± *0.62*	2.01 ± *0.03* ^e^	nd	nd
*B. racemosa*	0.08 ± *0.01* ^a^	34.30 ± *0.28* ^f^	53.71 ^c^ ± *0.50*	0.35 ± *0.01* ^b^	0.45 ± *0.04* ^ab^	0.06 ± *0.01* ^a^
*C. fistula*	0.06 ± *0.01* ^a^	34.21 ± *0.94* ^ef^	54.51 ^cd^ ± *0.69*	0.30 ± *0.03* ^ab^	0.46 ± *0.02* ^ab^	0.06 ± *0.01* ^a^
*D. latifolia*	0.11 ± *0.01* ^a^	22.44 ± *0.81* ^c^	47.35 ^b^ ± *1.23*	2.35 ± *0.09* ^e^	0.29 ± *0.03* ^a^	nd
*D. regia*	0.34 ± *0.03* ^b^	14.57 ± *0.32* ^a^	56.99 ^e^ ± *0.71*	0.61 ± *0.02* ^c^	nd	nd
*E. phaseoloides*	0.06 ± *0.01* ^a^	24.93 ± *0.36* ^d^	59.18 ^f^ ± *0.91*	0.57 ± *0.14* ^bc^	0.34 ± *0.06* ^a^	nd
*H. binata*	nd	19.28 ± *0.31* ^b^	40.78 ^a^ ± *1.43*	0.21 ± *0.12* ^a^	2.41 ± *0.05* ^c^	0.34 ± *0.02* ^c^
*P. pterocarpum*	0.61 ± *0.03* ^c^	19.37 ± *0.43* ^b^	46.75 ^b^ ± *0.91*	0.22 ± *0.04* ^a^	nd	nd
*S. catechu*	0.54 ± *0.02* ^c^	32.04 ± *0.35* ^e^	42.34 ^a^ ± *1.16*	0.25 ± *0.01* ^a^	0.35 ± *0.01* ^a^	nd
*S. sesban*	0.06 ± *0.02* ^a^	32.89 ± *0.53* ^e^	54.60 ^cd^ ± *0.62*	0.44 ± *0.03* ^b^	0.35 ± *0.02* ^a^	0.06 ± *0.01* ^a^
*V. nilotica*	nd	26.20 ± *0.49* ^d^	52.39 ^c^ ± *0.51*	1.02 ± *0.03* ^d^	0.25 ± *0.02* ^a^	nd

Values are expressed as the mean ± *standard deviation* (italic values) (*n* = 3). Different letters in the same column indicate statistically significant differences at *p* ≤ 0.05; nd, not detected. The following fatty acids were detected only in some of the species: C14:1 in *A. auriculiformis* (0.67%), *A. lebbeck* (0.08%) and *S. catechu* (0.17%); C22:1 in *D. latifolia* (0.26%); C22:2 in *A. auriculiformis* (0.15%), *A. lebbeck* (0.13%) and *H. binata* (0.93%).

**Table 3 molecules-28-03994-t003:** Carotenoid, tocopherol and tocotrienol content (mg/100 g oil) in the investigated seed oils.

Species	Total Carotenoids	Tocohromanols							
		α-T	β-T	γ-T	δ-T	α-T3	γ-T3	δ-T3	Total
*A. auriculiformis*	23.74 ± *1.39* ^f^	82.46 ± *2.80* ^e^	5.98 ± *0.24* ^b^	16.41 ± *0.70* ^a^	2.11 ± *0.30* ^b^	0.79 ± *0.04* ^b^	1.13 ± *0.05* ^b^	0.47± *0.03* ^a^	109.36 ± *4.17* ^a^
*A. concinna*	3.03 ± *0.11* ^b^	98.76 ± *7.0* ^f^	2.62 ± *0.27* ^a^	24.58 ± *2.64* ^b^	0.83 ± *0.09* ^a^	nd	nd	0.77 ± *0.10* ^ab^	127.57 ± *9.26* ^b^
*A. lebbeck*	18.55 ± *1.09* ^e^	101.59 ± *6.67* ^f^	0.97 ± *0.10* ^a^	68.84 ± *3.41* ^d^	1.95 ± *0.14* ^b^	nd	nd	nd	173.34 ± *10.31* ^d^
*A. odoratissima*	20.37 ± *1.07* ^e^	142.60 ± *6.44* ^g^	2.40 ± *0.14* ^a^	141.87 ± *5.52* ^g^	5.87 ± *0.25* ^d^	0.10 ± *0.02* ^a^	0.34 ^a^ ± *0.05*	nd	293.18 ± *1.68* ^g^
*B. racemosa*	6.80 ± *0.28* ^c^	125.02 ± *7.19* ^g^	1.50 ± *0.31* ^a^	30.28 ± *2.21* ^b^	0.61 ± *0.06* ^a^	nd	nd	nd	157.41 ± *9.11* ^c^
*C. fistula*	8.57 ± *0.53* ^c^	173.66 ± *7.87* ^h^	9.84 ± *0.45* ^c^	44.74 ± *2.32* ^c^	18.62 ± *0.37* ^e^	nd	nd	0.77 ± *0.05* ^ab^	247.63 ± *8.30* ^f^
*D. latifolia*	3.92 ± *0.24* ^b^	2.24 ± *0.14* ^a^	nd	278.74 ± *12.33* ^h^	2.66 ± *0.13* ^c^	nd	nd	nd	283.64 ± *12.60* ^g^
*D. regia*	13.29 ± *0.86* ^d^	209.27 ± *11.71* ^i^	17.53 ± *0.92* ^d^	113.83 ± *7.68* ^f^	5.81 ± *0.32* ^d^	4.93 ± *0.15* ^c^	14.27 ± *0.36* ^c^	1.33 ± *0.07* ^b^	367.51 ± *3.42* ^h^
*E. phaseoloides*	0.72 ± *0.05* ^a^	9.90 ± *0.34* ^b^	nd	87.87 ± *3.57* ^e^	nd	0.51 ± *0.05* ^b^	1.53 ± *0.06* ^b^	0.43 ± *0.03* ^a^	100.32 ± *4.08* ^a^
*H. binata*	3.99 ± *0.26* ^b^	171.98 ± *6.94* ^h^	4.96 ± *0.20* ^b^	64.77 ± *2.57* ^d^	5.60 ± *0.22* ^d^	nd	nd	nd	247.31 ± *9.93* ^f^
*P. pterocarpum*	12.04 ± *0.49* ^d^	160.18 ± *6.34* ^h^	1.04 ± *0.06* ^a^	84.66 ± *0.68* ^e^	2.50 ± *0.06* ^c^	nd	nd	nd	248.39 ± *6.73* ^f^
*S. catechu*	12.57 ± *0.89* ^d^	94.78 ± *1.86* ^f^	0.68 ± *0.05* ^a^	38.98 ± *1.81* ^c^	0.91 ± *0.03* ^a^	0.63 ± *0.04* ^b^	1.08 ± *0.10* ^b^	0.30 ± *0.03* ^a^	137.36 ± *3.93* ^b^
*S. sesban*	23.57 ± *1.78* ^f^	65.91 ± *3.26* ^d^	0.52 ± *0.13* ^a^	142.13 ± *3.95* ^g^	3.02 ± *0.12* ^c^	nd	0.46 ± *0.05* ^a^	nd	212.04 ± *7.01* ^e^
*V. nilotica*	3.28 ± *0.28* ^b^	43.98 ± *1.84* ^c^	2.14 ± *0.32* ^a^	76.46 ± *3.13* ^de^	3.86 ± *0.28* ^cd^	nd	0.39 ± *0.08* ^a^	nd	126.82 ± *5.12* ^b^

Values are expressed as the mean ± *standard deviation* (italic values) (*n* = 3). Different letters in the same column indicate statistically significant differences at *p* ≤ 0.05. In *D. regia*, β-T3 was detected (0.5 mg/100 g oil). T, tocopherol; T3, tocotrienol; nd, not detected.

**Table 4 molecules-28-03994-t004:** Sterol and squalene content (mg/100 g oil) in the seed oils.

Species	Sterols													Squalene
	Cholesterol	Campesterol	Campestanol	Δ5-Stigmasterol	β-Sitosterol	Sitostanol	Δ5-Avenasterol	Cycloartenol	Cycloeucalenol	24Methyl-zymosterol	Δ7-Avenasterol	Δ7-Stigmasterol	24-Methylene-cycloarterol	
*A. auriculiformis*	nd	nd	nd	8 ± *0* ^a^	211 ± *9* ^b^	nd	6 ± *0* ^a^	25 ± *1* ^b^	nd	88 ± *3* ^b^	nd	nd	nd	nd
*A. concinna*	nd	70 ± *4* ^c^	6 ± *2* ^a^	59 ± *8* ^b^	1306 ± *156* ^g^	nd	313 ± *13* ^d^	207 ± *6* ^f^	136 ± *7* ^d^	nd	nd	446 ± *15* ^d^	nd	nd
*A. lebbeck*	nd	14 ± *1* ^a^	11 ± *1* ^b^	13 ± *2* ^a^	295 ± *10* ^c^	7 ± *1* ^a^	27 ± *1* ^b^	26 ± *2* ^b^	nd	nd	7 ± *0* ^a^	nd	nd	nd
*A. odoratissima*	nd	35 ± *3* ^b^	5 ± *1* ^a^	23 ± *1* ^a^	380 ± *13* ^d^	nd	32 ± *3* ^b^	24 ± *1* ^b^	nd	nd	nd	nd	nd	nd
*B. racemosa*	nd	69 ± *2* ^c^	nd	155 ± *9* ^c^	520 ± *18* ^e^	nd	nd	6 ± *0* ^a^	4 ± *0* ^a^	nd	44 ± *4* ^c^	50 ± *5* ^b^	7 ± *1* ^a^	31 ± *2* ^a^
*C. fistula*	nd	86 ± *4* ^d^	nd	249 ± *18* ^e^	854 ± *24* ^f^	6 ± *1* ^a^	27 ± *3* ^b^	44 ± *3* ^c^	67 ± *5* ^c^	nd	nd	nd	11 ± *1* ^a^	303 ± *9* ^b^
*D. latifolia*	nd	36 ± *2* ^b^	11 ± *1* ^b^	13 ± *1* ^a^	291 ± *12* ^c^	3 ± *0* ^a^	23 ± *2* ^b^	11 ± *1* ^a^	nd	nd	6 ± *1* ^a^	nd	nd	nd
*D. regia*	49 ± *3* ^c^	142 ± *6* ^e^	11 ± *1* ^b^	209 ± *9* ^d^	502 ± *16* ^e^	34 ± *4* ^b^	28 ± *3* ^b^	129 ± *7* ^e^	nd	nd	nd	72 ± *3* ^b^	9 ± *1* ^a^	nd
*E. phaseoloides*	nd	11 ± *1* ^a^	nd	14 ± *1* ^a^	133 ± *5* ^b^	nd	30 ± *2* ^b^	29 ± *1* ^b^	nd	nd	16 ± *2* ^ab^	nd	8 ± *1* ^a^	nd
*H. binata*	nd	129 ± *5* ^e^	10 ± *1* ^b^	53 ± *3* ^b^	390 ± *17* ^d^	17 ± *1* ^a^	46 ± *4* ^c^	29 ± *2* ^b^	nd	nd	48 ± *3* ^c^	nd	8 ± *1* ^a^	nd
*P. pterocarpum*	18 ± *1* ^a^	66 ± *3* ^c^	2 ± *1* ^a^	273 ± *11* ^e^	11 ± *1* ^a^	107 ± *5* ^c^	2 ± *0* ^a^	63 ± *5* ^d^	nd	nd	nd	90 ± *4* ^c^	13 ± *1* ^a^	nd
*S. catechu*	nd	265 ± *9* ^f^	nd	598 ± *24* ^f^	396 ± *15* ^d^	nd	27 ± *2* ^b^	40 ± *2* ^c^	nd	25 ± *2* ^a^	nd	nd	nd	nd
*S. sesban*	33 ± *2* ^b^	109 ± *4* ^d^	nd	301 ± *9* ^e^	354 ± *14* ^cd^	10 ± *1* ^a^	34 ± *2* ^b^	31 ± *2* ^b^	27 ± *3* ^b^	nd	25 ± *2* ^b^	nd	nd	41 ± *2* ^a^
*V. nilotica*	nd	68 ± *3* ^c^	2 ± *0* ^a^	4 ± *0* ^a^	592 ± *17* ^e^	nd	33 ± *2* ^b^	25 ± *2* ^b^	nd	nd	nd	12 ± *1* ^a^	nd	44 ± *3* ^a^

Values are expressed as the mean ± *standard deviation* (italic values) (*n* ± 3). Different letters in the same column indicate statistically significant differences at *p* ≤ 0.05; nd, not detected. The following sterols were detected only in some of the species and expressed in mg/100 g oil (values are provided in brackets): α-amyrin in *C. fistula* (58) and *S. sesban* (60); brassicasterol in *D. regia* (19) and *P. pterocarpum* (24); clerosterol in *A. odoratissima* (18) and *P. pterocarpum* (24); gramisterol in *A. odoratissima* (30), *D. regia* (66) and *P. pterocarpum* (64); Δ5,23-stigmastadienol and 24-methylene cholesterol in *P. pterocarpum* (26 and 48, respectively); Δ5,24-stigmastadienol in *A. odoratissima* (17).

**Table 5 molecules-28-03994-t005:** Plant type and common uses of the investigated species.

Species	Plant Type	Use						
		Timber	Agroforestry	Ornamental	Traditional Medicines	Fodder	Food	Fuelwood
*A. auriculiformis*	Tree	e	a b c h i*	c		a c		a c
*A. concinna*	Tree		i*		o		•	•
*A. lebbeck*	Tree	c	c	c	f o	c g h		c
*A. odoratissima*	Tree	q	q		f j	•		
*B. racemosa*	Tree			k	k	k		
*C. fistula*	Tree	e		p	f o	•	•	
*D. latifolia*	Tree	e						
*D. regia*	Tree		a			a	a	a
*E. phaseoloides*	Liana				m n		m	
*H. binata*	Tree	e				•		
*P. pterocarpum*	Tree			p		g		
*S. catechu*	Tree		i*		f		d	
*S. sesban*	Tree		h i*			l	•	l
*V. nilotica*	Tree	e	h i*		•	h	•	

Letters in the table denote references: a [4]; b [1]; c [5]; d [47]; e [9]; f [48]; g [10]; h [7]; i [6]*—only genus mentioned; j [49]; k [50]; l [11]; m [51]; n [52]; o [53]; p [54]; q [55]; • mentioned in online databases with no reference provided.

## Data Availability

The data used to support the findings of this study are available in Appendix A and from the corresponding author upon request.

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
