# Peer review of "Evaluation of Selected Medicinal, Timber and Ornamental Legume Species’ Seed Oils as Sources of Bioactive Lipophilic Compounds"

_molecules, 2023, doi:10.3390/molecules28103994_

Round 1
Reviewer 1 Report
The manuscript concerns an essential topic of ‘Evaluation of selected medicinal, timber, and ornamental legume species’ seed oils as sources of bioactive lipophilic compounds. The research is interesting and has novelty. However, It has been seen that the authors have dealt with the issues very quickly. Where need more focus on the manuscript. And it needs extensively revised. Moreover, linguistically, the manuscript needs to revise by the native expert. There are some points that should be addressed before the manuscript is suitable for publication in ‘Molecules’.
Abstract
The abstract needs 1 or 2 lines introduction about the study. You need to display in the abstract why you want to do this study.
Line 2 to line 3 in the abstract; all specie name should be italic
Question1: Why the authors chose this tree type of plants (ornamental, medicinal, agroforestry, or timber)? Why didn't you choose vegetable plants, non-timber plants, or crops or fruits? Please mention this in the manuscript.
One standard abstract should include an introduction, material methods, Results, and discussion. This abstract need to rewrite again to reach the standard of the journal . For example, you need to start with the result shown…… We concluded.......
All data percentages should be shown with two or one digital number.
The keywords should not be the same word in the title and abstract. Please check and replace it with other keywords.
Introduction
The structure of the introduction is confusing. The introduction should be rewritten again. The aim of this study should be at the end of the introduction part. What is the author's hyposis? Why do you use the table in the introduction? Don't use long sentences. Use more citations in sentences. The paragraph location is not suitable, please rewrite carefully.
Introduction need to start with this paragraph ‘’Plant parts of several species are used in food, such as the fresh leaves of A. auriculi……..
This paragraph should move to the discussion part or you can use only first line in introduction. species name should use in material and methods ’’ In the present study, the seeds of 14 timber, ornamental or medicinal legume species were analysed for their lipid profile: Acacia auriculiformis (auri), Senegalia catechu = Acacia catechu (catechu), Acacia concinna = Senegalia rugata (shikakai), Vachellia nilotica = Acacia nilotica (Gum Arabic tree), Albizia lebbeck (siris, sirisa), Albizia odoratissima (Ceylon rosewood, black siris), Bauhinia racemosa (Bidi-leaf tree), Cassia fistula (golden shower), Dalbergia latifolia (Indian rosewood), Delonix regia (Royal poinciana), Entada phaseoloides (box bean), Hardwickia binata (anjan), Peltophorum pterocarpum (copperpod) and Sesbania sesban (Egyptian riverhemp, sesban)’’.
Table 1 can be moved to material and methods.
Use these new references in the introduction parts
1. Sun, J., Jia, Q., Li, Y., Zhang, T., Chen, J., Ren, Y.,... Fu, S. (2022). Effects of Arbuscular Mycorrhizal Fungi and Biochar on Growth, Nutrient Absorption, and Physiological Properties of Maize (Zea mays L.). Journal of Fungi, 8(12). doi: 10.3390/jof8121275
2. Yang, K., Geng, Q., Luo, Y., Xie, R., Sun, T., Wang, Z.,... Tian, J. (2022). Dysfunction of FadA-cAMP signalling decreases Aspergillus flavus resistance to antimicrobial natural preservative Perillaldehyde and AFB1 biosynthesis. Environmental microbiology, 24(3), 1590-1607. doi: 10.1111/1462-2920.15940
3. Pan, C., Yang, K., Erhunmwunsee, F., Li, Y., Liu, M., Pan, S.,... Tian, J. (2023). Inhibitory effect of cinnamaldehyde on Fusarium solani and its application in postharvest preservation of sweet potato. Food Chemistry, 408, 135213. doi: https://doi.org/10.1016/j.foodchem.2022.135213
4. Zhang, Y., Zhang, S., Yang, X., Wang, W., Liu, X., Wang, H.,... Zhang, H. (2022). Enhancing the fermentation performance of frozen dough by ultrasonication: Effect of starch hierarchical structures. Journal of Cereal Science, 106, 103500. doi: https://doi.org/10.1016/j.jcs.2022.103500
5. Wang, Y., Liu, S., Yang, X., Zhang, J., Zhang, Y., Liu, X.,... Wang, H. (2022). Effect of germination on nutritional properties and quality attributes of glutinous rice flour and dumplings. Journal of Food Composition and Analysis, 108, 104440. doi: https://doi.org/10.1016/j.jfca.2022.104440
6. Li, M., Xia, Q., Lv, S., Tong, J., Wang, Z., Nie, Q.,... Yang, J. (2022). Enhanced CO2 capture for photosynthetic lycopene production in engineered Rhodopseudomonas palustris, a purple nonsulfur bacterium. Green chemistry, 24(19), 7500-7518. doi: 10.1039/d2gc02467e
Materials and methods
This part needs to start with 2.1 plant material. Please replace this paragraph.
In the plant material section from lines 1- 5 all specie name should be italic
In the section of plant material line 7 . Species authentication was done by prof. Khageshwar Singh Patel using a standard monograph. Please use ‘’by the method of Singh Patel by using a standard monograph’’. and cited him.
2-3. ‘using ultrasound-assisted extraction’ this part should move to the text.
In Total carotenoid analysis for the definition of the letters that's better to use the equal symbol. (=) such as C= A= …..
In Statistical analysis, please exactly mention how many replications the authors used for each treatment.
Results and discussion
The author could show some indexes with some figures. The tables are not standard, the authors can divide them into two tables or some figures. The discussion is insufficient. Please use more citations' of literature.
This sentence should move to introduction ‘To the best of our knowledge, this study includes the first investigation of D. latifolia…..
Table 2 can divide into two tables, or the author can show with figures. Please revise it.
Mean difference letters should move here 8.98±0.18bc
Conclusions
Conclusions is too short please add some more details

Author Response
Thank you for your comments, the English has been improved.
Thank you for your comments, the right sentence has been added.
All Latin names of the species are provided in italic script in the submitted manuscript. Words not given in italic script are the common names of the species. These can be the anglicized word used for the plant in the areas it is native to and can coincide with the Latin name in some cases. The common names for plants are not usually italicized and are therefore provided in straight script. To avoid confusion it the future, this comment, shortened and paraphrased, has been included in the manuscript.
Thank you for the comment. Tree legumes were investigated due to the relatively scarce information on their lipophilic profile. This, paraphrased, has been added to the manuscript.
The abstract has been revised to fit a more standard format.
Thank you for the comment. The numerical values provided in the text have been corrected to include two significant figures after the decimal point.
Thank you for the comment, the key words have been amended so that none of the key words used appear in the title or abstract.
Thank you for the suggestions, the introduction has been partly rewritten and sentences moved to paragraphs of highest relevance. Due to the exploratory nature of the study, it is not possible to include a hypothesis that follows a traditional format of “if a, then b” or “y is dependent on x”, instead, the study implies the potential of understudied species as sources of bioactive compounds as a hypothesis. Sentences have been edited to more standard phrasing. Much of the information in the introduction can be considered general knowledge and does not require a reference.
We thank you for the suggestion, but the species included in the study are not primarily used for food.
Thank you for the suggestion, the species’ Latin and common names were provided in the beginning of the introduction to alleviate looking up the full name of the plant if needed. The listing of investigated species has been moved to the “Plant material” section of the methodology.
Thank you for the suggestion, the Table 1 has been moved to material and methods.
Although we thank you for the suggestions of additional literature, they do not relate to the topics of the study. The present study was not related to microbiology. However, we have included the references from the provided list that fit the topic of the present study.
The paragraph on plant material has been moved to the beginning of the section.
The Latin names of all species are italicized in the text. The names given in straight script are common names of the species. Although many of these are not native to English, the untranslated word has been adopted in English and was therefore not italicized.
Thank you for the suggestion, the sentence has been reworded slightly.
It is not entirely clear which text the phrase should be moved to, it has been removed from the beginning of the section and integrated into the appropriate part of the method description.
Explanation of letter and symbols in formulas have been amended.
Thank you for the comment, the number of replications and their nature has been made more explicit in the text.
Thank you for the suggestion, figures have been added to the manuscript where they are appropriate.
The sentence, reworded, has been moved to the introduction.
Thank you for the suggestion, Table 2 has been divided into two tables, however, showing the results in a figure is difficult due to the very different values (they differ more than 10 times in many cases) and large number of analytes (grayscale and colored charts are illegible).
Thank you for the comment, mean difference numbers have been moved in all tables.
The conclusions have been supplemented.
Reviewer 2 Report
Please see comments in the attached file.

Author Response
All species names are written in italic in the original manuscript.
Thank you, this has been addressed in the revised abstract.
Thank you for the comment. A short justification for the investigation of lipophilic bioactive compounds has been added to the introduction.
All species Latin names are written in italic in the original manuscript. Common names are written in straight script. To avoid future misunderstanding, the choice of italic/straight script has been explained where the common names appear.
Thank you for the comment, the mentioned examples have been corrected as well as any other such instances in the manuscript.
Thank you for the suggestion, the text has been supplemented with some possible reasons for lower oil yield.
Thank you for the suggestion, some more recent references have been added.
Reviewer 3 Report
The manuscript aims to evaluate some chemical lipophilic components from the seeds of 14 legume species of importance in the ornamental, medicinal, agroforestry or timber fields, for their possible use as food.
After extraction of the oil from the seeds, the authors evaluated the composition of saturated and unsaturated fatty acids, carotenoids, tochromanols and sterols in each variety.
Saturated and unsaturated fatty acids were determined by GC and GC-MS analysis, carotenoids were detected by UV spectrophotometric analysis, and tocochromanols were identified by HPLC analysis with a fluorescence detector. GC-MS analysis was also used to detect sterols.
General comments:
I believe that for the topics covered the manuscript is closer to the aims and scope of “Horticulturae” or “Agronomy” journal rather than “Molecules”.
The topic is interesting and well presented, however the authors should include in the supplementary material: the GC-MS profile, the HPLC chromatogram, the UV spectrum for the identified compounds. In the GC-MS analysis of sterols, the Rt and Kovats retention index must be reported.
The manuscript can be accepted after major revisions.
Author Response
Thank you for the comment. Well, this is a debatable topic. Any aspects associated with the horticulture and agronomy topics were not investigated in the present study. We focused on bioactive molecules present in the poorly investigated species of the Fabaceae family.
Thank you for the positive overview. Fatty acids, sterols, and squalene were quantified by GC-FID, while GC-MS and Kovats retention index are used only in not clear issues. Tocopherols and tocotrienols were detected by HPLC-FLD, while total carotenoids were by spectrophotometer. In the supplementary file (excel file) the HPLC-FLD chromatograms of tocopherols and tocotrienols determination, the GC-FID chromatograms of the fatty acids, sterols, and squalene, and Kovats retention index were provided. Were not able to provide the spectra of carotenoids measurements due to technical issues - we apologize for this.
Thank you for your positive recommendation.
Round 2
Reviewer 1 Report
It has been obvious that the authors improved the quality of the paper. I think the paper is suitable to be accepted by the journal.
Reviewer 3 Report
The manuscript can be accepted in the present form